# Weight-Entanglement Meets Gradient-Based Neural Architecture Search

**Rhea Sanjay Sukthanker[1], Arjun Krishnakumar[1], Mahmoud Safari[1], Frank Hutter[2,1]**

[1]University of Freiburg, [2]ELLIS Institute Tübingen
`{sukthank,krishnan,safarim,fh}@cs.uni-freiburg.de`

**Abstract** Weight sharing is a fundamental concept in neural architecture search (NAS), enabling gradient-based methods to explore cell-based architectural spaces significantly faster than traditional black-box approaches. In parallel, weight-*entanglement* has emerged as a technique for more intricate parameter sharing amongst macro-architectural spaces. Since weight-entanglement is not directly compatible with gradient-based NAS methods, these two paradigms have largely developed independently in parallel sub-communities. This paper aims to bridge the gap between these sub-communities by proposing a novel scheme to adapt gradient-based methods for weight-entangled spaces. This enables us to conduct an in-depth comparative assessment and analysis of the performance of gradient-based NAS in weight-entangled search spaces. Our findings reveal that this integration of weight-entanglement and gradient-based NAS brings forth the various benefits of gradient-based methods, while preserving the memory efficiency of weight-entangled spaces. The code for our work is openly accessible here.

## 1 Introduction

The concept of weight-sharing in Neural Architecture Search (NAS) was motivated by the need to improve efficiency over that of conventional black-box NAS algorithms, which demand significant computational resources to evaluate individual architectures. Here, weight-sharing (WS) refers to the paradigm by which we represent the search space with a single large *supernet*, also known as the *one-shot* model, that subsumes all the candidate architectures in that space. Every edge of this supernet holds all the possible operations that can be assigned to that edge. Importantly, architectures that share a particular operation also share its corresponding operation weights, allowing for efficient simultaneous partial training of an exponential number of subnetworks with each gradient update.

Gradient-based NAS algorithms (or *optimizers*), such as DARTS (Liu et al., 2019), GDAS (Dong and Yang, 2019) and DrNAS (Chen et al., 2021b), assign an *architectural parameter* to every choice of operation on a given edge of the supernet. The output feature maps of these edges are thus an aggregation of the outputs of the individual operations on that edge, weighted by their architectural parameters. These architectural parameters are learned using gradient updates by differentiating through the validation loss. Supernet weights and architecture parameters are therefore trained simultaneously in a bi-level fashion. Once this training phase is complete, the final architecture can be obtained quickly, e.g., by selecting operations with the highest architectural weights on each edge as depicted in Figure 1(b). However, more sophisticated methods have also been explored (Wang et al., 2021a) for this selection.

While gradient-based NAS methods have primarily been studied for cell-based search spaces, a different class of search spaces focuses on macro-level structural decisions, such as the number of channels in a layer of the supernet, or the number of layers which are stacked to form the supernet. In these spaces, all architectures are *subnetworks* of the architecture with the largest architectural choices, identical to the supernet in this case. These search spaces share weights more intricately

---

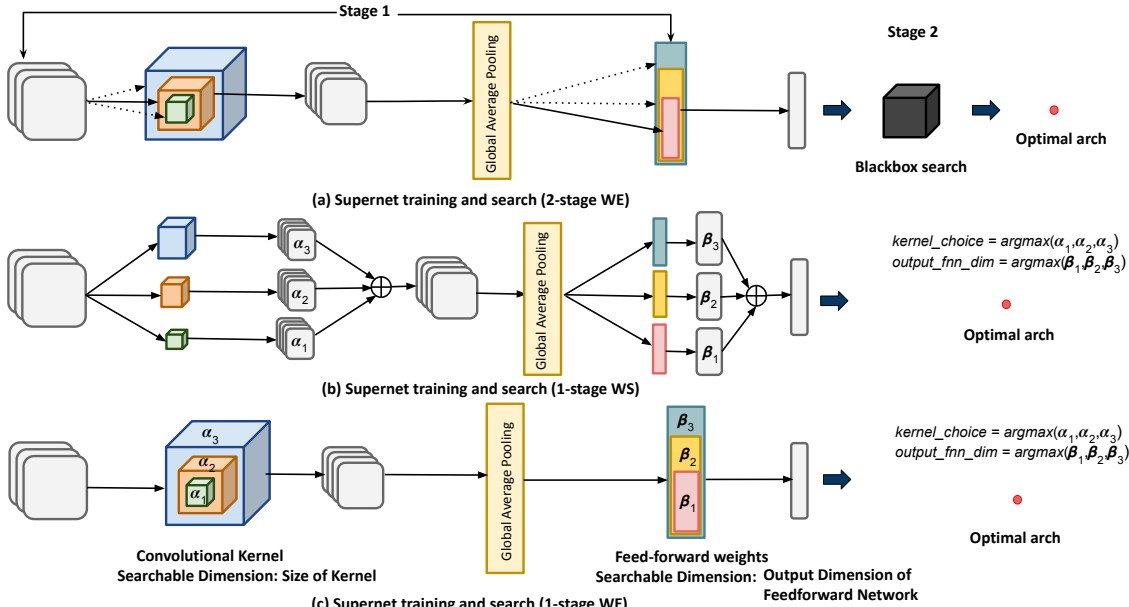

Figure 1: (a) **Two-Stage NAS with WE** (Algorithm 3): dotted paths show operation choices not sampled at the given step (b) **Single-Stage NAS with WS** (Algorithm 4): every operation choice is evaluated independently and contributes to the output feature map with corresponding architecture parameters (c) **Single-Stage NAS with WE** (Algorithm 1): operation choices superimposed with corresponding architecture parameters. The architecture parameters for the three operation choices are represented by $[\alpha_i]_{i=1}^3$ and $[\beta_i]_{i=1}^3$. The operation weights, or choices, are symbolized by cubes (for convolutional layers) or rectangles (for feedforward layers) in various colors. In scenarios (a) and (c), due to weight entanglement, the smaller weights are effectively structured subsets of the larger weights. Conversely, in (b), through weight-sharing, operation weights are maintained independently from each other. In both (b) and (c), to determine the optimal architecture, the operations associated with the highest architecture parameter value are selected. This selection process applies to the choice of kernel size and the output dimension of the feedforward network.

via *weight-entanglement* (WE) between similar operations on the same edge. An example of this for convolutional layers is that the 9 weights of a $3 \times 3$ convolution are a subset of the 25 weights of a $5 \times 5$ convolution. This paradigm reduces the memory footprint of the supernet to the size of the largest architecture in the space, unlike WS search spaces, where it increases linearly with the number of operation choices.

In order to efficiently search over such weight-entanglement spaces, *two-stage* methods first pre-train the supernet and then perform black-box search on it to obtain the final architecture. OFA (Cai et al., 2020), SPOS (Guo et al., 2020), AutoFormer (Chen et al., 2021a) and HAT (Wang et al., 2020) are prominent examples of methods that fall into this category. Note that these methods do not employ additional architectural parameters for supernet training or search. They typically train the supernet by randomly sampling subnetworks and training them as depicted in Figure 1(a). The post-hoc black-box search relies on using the performance of subnetworks sampled from the trained supernet as a proxy for true performance on the unseen test set. To contrast with this two-stage approach, we refer to traditional gradient-based NAS approaches as *single-stage* methods.

To date, weight-entangled spaces have only been explored with two-stage methods, and cell-based spaces only with single-stage approaches. In this paper, we bridge the gap between these parallel sub-communities. We do so by addressing the challenges associated with integrating off-the-shelf single-stage NAS methods with weight-entangled search spaces. After a discussion of related work (Section 2), we make the following main contributions:

1. We propose a generalized scheme to apply single-stage methods to weight-entangled spaces while maintaining search **efficiency** and **efficacy** at larger scales (Section 3, with visualizations in Figure 1(c) and Figure 2 and Figure 3). We refer to this method as *TangleNAS*.

2. We propose a **unified evaluation framework** for the comparative evaluation of single and two-stage methods (Section 4.1) and study the effect of weight-entanglement in conventional cell-based search spaces (i.e., NAS-Bench-201 and the DARTS search space) (Section 4.2).

3. We evaluate our proposed generalized scheme for single-stage methods across a **diverse set of weight-entangled macro search spaces and tasks**, from image classification (Section 4.3.1 and Section 4.3.2) to language modeling (Section 4.3.3).

4. We conduct a comprehensive evaluation of the properties of single and two-stage approaches (Section 5), demonstrating that our generalized gradient-based NAS method **achieves the best of single and two-stage methods**: the enhanced performance, improved supernet fine-tuning properties, superior any-time performance of single-stage methods, and low memory consumption of two-stage methods. To facilitate reproducibility, our code is openly accessible here.

## 2 Related Work

*Weight-sharing* was first introduced in ENAS (Pham et al., 2018), which reduced the computational cost of NAS by 1000× compared to previous methods. However, since this method used reinforcement learning, its performance was quite brittle. Bender et al. (2018) simplified the technique, showing that searching for good architectures is possible by training the supernet directly with stochastic gradient descent. This was followed by DARTS (Liu et al., 2019), which set the cornerstone for efficient and effective gradient-based, *single-stage* NAS approaches.

DARTS, however, had prominent *failure modes*, such as its *discretization gap* and *convergence towards parameter-free operations* (White et al., 2023), as outlined in Robust-DARTS (Zela et al., 2020). Numerous gradient-based one-shot optimization techniques were developed since then (Cai et al., 2019; Nayman et al., 2019; Wang et al., 2021b; Dong and Yang, 2019; Hu et al., 2020; Li et al., 2021; Chen et al., 2021b; Zhang et al., 2021). Amongst these, we highlight DrNAS (Chen et al., 2021b), which we will use in our experiments as a representative of gradient-based NAS methods. DrNAS treats one-shot search as a distribution learning problem, where the parameters of a Dirichlet distribution over architectural parameters are learned to identify promising regions of the search space. Despite the remarkable performance of single-stage methods, they are not directly applicable to some real-world architectural domains, such as transformers, due to the macro-level structure of these search spaces. DASH (Shen et al., 2022) employs a DARTS-like method to optimize CNN topologies (i.e., kernel size, dilation) for a diverse set of tasks, reducing computational complexity by appropriately padding and summing kernels with different sizes and dilations. FBNet-v2 (Wan et al., 2020) and MergeNAS (Wang et al., 2021b) make an attempt along these lines for CNN topologies, but their methodology is not easily extendable to search spaces like transformers with multiple interacting modalities, such as embedding dimension, number of heads, expansion ratio, and depth.

Weight-entanglement, on the other hand, provides a more effective way of weight-sharing, exclusive to macro-level architectural spaces. In weight-entanglement, operations with weights of smaller dimensionality are structured subsets of the largest dimension. Hence, the total number of parameters in the supernet is the same as the number of parameters in the largest architecture in the space. The concept of weight-entanglement was developed in slimmable networks (Yu et al., 2018; Yu and Huang, 2019), OFA (Cai et al., 2020) and BigNAS (Yu et al., 2020) in the context of convolutional networks (see also AtomNAS (Mei et al., 2019)) and later spelled out in AutoFormer (Chen et al., 2021a) and applied to the transformer architecture.

Single-path-one-shot (SPOS) methods (Guo et al., 2020) have shown a lot of promise in searching weight-entangled spaces. SPOS trains a supernet by uniformly sampling paths (one at a time to limit memory consumption) and then training the weights along that path. The supernet

training is followed by a black-box search that uses the performance of the models sampled from the trained supernet as a *proxy*. OFA used a similar idea to optimize different dimensions of CNN architectures, such as its depth, width, kernel size, and resolution. Additionally, it enforced training of larger to smaller subnetworks sequentially to prevent interference between subnetworks. Subsequently, AutoFormer adopted the SPOS method to optimize a weight-entangled space of transformer architectures.

In this work, we demonstrate the application of single-stage methods to macro-level search spaces with weight-entanglement. This approach leverages the time efficiency and effectiveness of modern differentiable NAS optimizers, while maintaining the memory efficiency inherent to the weight-entangled space. Although DrNAS is our primary method for exploring weight-entangled spaces, our methodology is broadly applicable to other gradient-based NAS methods.

## 3 Methodology: Single-Stage NAS with Weight-Superposition

**Computational Efficiency**. Our primary goal in this work is to effectively apply single-stage NAS to search spaces with macro-level architectural choices. To reduce the memory consumption and preserve computational efficiency, we propose two major modifications to single-stage methods. Firstly, the weights of the operations on

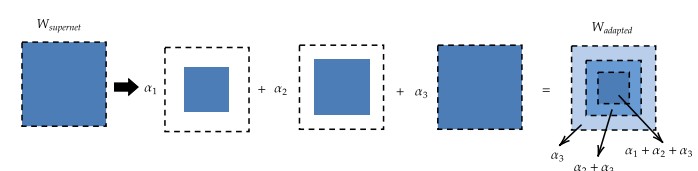

Figure 2: Weight superposition with architecture parameters $\alpha_i{}_{i=1}^{3}$ for kernel size search. Supernet weight matrix (LHS) is adapted to gradient-based methods (RHS).

every edge are shared with the corresponding weights of the largest operator on that edge. This reduces the size of the supernet to the size of the largest individual architecture in the search space.

Secondly, to compute operation mixture, we take each operation choice, zero-pad to match the size of the largest choice, and then sum these, with each one multiplied by its respective architectural parameter. The resulting operation thus represents a mixture of different choices on an edge. This approach, visualized in Figure 1(c), contrasts with single-stage NAS methods, which weigh the *out-*

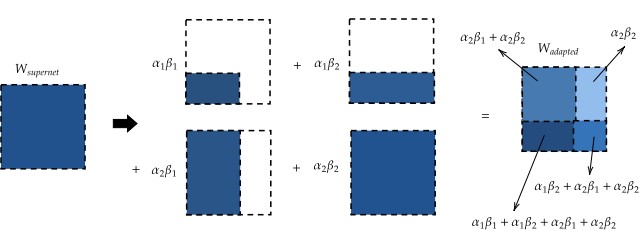

Figure 3: Combi-superposition with parameters $\alpha_i\beta_j$. Supernet weight matrix (LHS) is adapted to gradient-based methods (RHS).

*puts* of operations on a specific edge using architectural parameters (refer to Figure 1(b)). Figure 2 provides an overview of the idea for a single architectural choice, such as the kernel size. This is equivalent to taking the largest operation and re-scaling the weights of each sub-operation by corresponding architectural parameters followed by summation of the weights (see the right-most weight matrix in Figure 2).

**Weight Superposition**. *Weight Superposition* is defined as a weighted summation of subsets of the largest weight matrix, to obtain weights with structural properties comparable to the largest operation. Consequently, a *single* forward pass on the superimposed weights suffices to capture the effect of all operational choices, thus making our approach computationally efficient. See Figure 2 for details on weight superposition of the kernel size.

**Algorithm 1** TangleNAS

1: **Input**: $M \leftarrow$ number of cells, $N \leftarrow$ number of operations
$\quad\quad \mathcal{O} \leftarrow [o_1, o_2, o_3, ...o_N]$
$\quad\quad \mathcal{W}_{max} \leftarrow \cup_{i-1}^N w_i$
$\quad\quad \mathcal{A} \leftarrow [\alpha_1, \alpha_2, \alpha_3, ...\alpha_N]$
$\quad\quad \gamma$ = learning rate of $\mathcal{A}$, $\eta$ = learning rate of $\mathcal{W}_{max}$
$\quad\quad f$ is a function or distribution s.t. $\sum_{i=1}^N f(\alpha_i) = 1$
2: $Cell_j \leftarrow DAG(\mathcal{O}_j, \mathcal{W}_{max\,j})$ /* defined for j=1...M */
3: $Supernet \leftarrow \cup_j^M Cell_j \cup \mathcal{A}$
4: /* example of forward propagation on the cell */
5: **for** $j \leftarrow 1$ to $M$ **do**
6: $\quad$ /* PAD weight to output dimension of $\mathcal{W}_{max}$ before summation */
7: $\quad$ /* Generalized Weighing Scheme */
8: $\quad$ $\overline{o_j(x, \mathcal{W}_{max}) = o_{j,i}(x, \sum_{i=1}^N f(\alpha_i)\,\mathcal{W}_{max}[: i])}$
9: **end for**
10: /* weights and architecture update */
11: $\mathcal{A} = \mathcal{A} - \gamma \nabla_{\mathcal{A}} \mathcal{L}_{val}(\mathcal{W}_{max}{}^*, \mathcal{A})$
12: $\mathcal{W}_{max} = \mathcal{W}_{max} - \eta \nabla_{\mathcal{W}_{max}} \mathcal{L}_{train}(\mathcal{W}_{max}, \mathcal{A})$
13: /* Architecture Selection */
14: $selected\_arch \leftarrow \arg\max(\mathcal{A})$

**Combi-Superposition**. An operation may depend on two or more architectural decisions. Consider, for example, embedding dimension and intermediate MLP dimension (where the latter depends on the former by a searchable multiplicative factor i.e., the MLP ratio). To accommodate this, we introduce combi-superposition outlined in Figure 3 and in Algorithm 2. Combi-superposition superimposes the weights across multiple different architectural dimensions, allowing for search across different interacting architectural modalities.

**Algorithm**. These concepts allow us to apply any arbitrary gradient-based NAS method, such as DARTS, GDAS, or DrNAS, to macro-level search spaces that leverage weight-entanglement without incurring additional memory and computational costs during forward propagation. We name this single-stage architecture search method *TangleNAS*. See Algorithm 1 for an overview of the bi-level optimization framework with weight-superposition. The operation $f$ in Algorithm 1 determines the differentiable optimizer used in the method. For example, $f$ is a *softmax* function for DARTS and a function that samples from the Dirichlet distribution for DrNAS. We use DrNAS as the primary gradient-based NAS method in all our experiments, and refer to DrNAS used in conjunction with weight-entangled spaces as *TangleNAS* in the remainder of the paper.

## 4 Experiments

We evaluate TangleNAS on a broad range of search spaces, from cell-based spaces (which serve as the foundation for single-stage methods) to weight-entangled convolutional and transformer spaces (which are central to two-stage methods). We initiate our studies by exploring two simple toy search spaces, which include a collection of cell-based and weight-entangled spaces. Later, we scale our experiments to larger analogs of these spaces. In all our experiments, we use *WE* to refer to the supernet type with entangled weights between operation choices and *WS* to refer to standard weight-sharing proposed in cell-based spaces. For details on our experimental setup, please refer to Appendix F. Furthermore, in all our experiments the focus is on *unconstrained search*, i.e., a scenario where the user is interested in obtaining the architecture with the best performance on their metric of choice, without constraints such as model size, or inference latency. The two-stage baselines we mainly compare against are SPOS (Guo et al., 2020) with Random Search (*SPOS+RS*) and SPOS with Evolutionary Search (*SPOS+ES*). For MobileNetV3 (Section 4.3.2) and ViT (Section 4.3.1), we use the original training pipeline from OFA (Cai et al., 2020) and Autoformer (Chen et al., 2021a), respectively, both of which use SPOS (Guo et al., 2020) as their foundation. However, for Once-for-All, we do not incorporate the progressive shrinking scheme during search.

### 4.1 Toy search spaces

We begin the evaluation of TangleNAS on two compact *toy* search spaces that we designed as a contribution to the community to allow faster iterations of algorithm development:

- **Toy cell space**: a small version of the DARTS space; architectures are evaluated on the Fashion-MNIST dataset (Xiao et al., 2017).

| Search Type | Optimizer | Supernet type | Test acc (%) | Search Time (hrs) |
|---|---|---|---|---|
| Single-Stage | DrNAS | WS | 91.190 ± 0.049 | 6.3 |
| | **TangleNAS** | WE | **91.300 ± 0.023** | 6.2 |
| Two-Stage | SPOS+RS | WE | 90.680 ± 0.253 | 15.6 |
| | | WE | 90.317 ± 0.223 | 13.2 |
| Optimum | - | - | 91.630 | - |

Table 1: Evaluation on the toy cell-based search space on the Fashion-MNIST dataset.

| Search Type | Optimizer | Supernet type | Test acc (%) | Search Time (hrs) |
|---|---|---|---|---|
| Single-Stage | DrNAS | WS | 10 ± 0.000 | 12.4 |
| | **TangleNAS** | WE | **82.495 ± 0.461** | 8.6 |
| Two-Stage | SPOS+RS | WE | 81.253 ± 0.672 | 21.7 |
| | | WE | 81.890 ± 0.800 | 26.4 |
| Optimum | - | - | 84.410 | - |

Table 2: Evaluation on the toy conv-macro search space on the CIFAR-10 dataset.

- **Toy conv-macro space**: a small space inspired by MobileNet, including kernel sizes and the number of channels in each convolution layer as architectural decisions. The architectures are evaluated on CIFAR-10.

We describe these spaces in Appendix D, including links to code for these open source toy benchmarks. The results of these experiments are summarized in Tables 1 and 2. In both of these search spaces, TangleNAS outperforms its *two-stage* counterparts over 4 seeds. Additionally, DrNAS without weight-entanglement performs extremely poorly on the macro level search space (equivalent to a random classifier).

## 4.2 Cell-based search spaces

We now begin our comparative analysis of single and two-stage approaches by applying them to cell-based spaces, which are central in the single-stage NAS literature. We evaluate TangleNAS against DrNAS and SPOS on these spaces. Here, we use the widely studied NAS-Bench-201 (NB201) (Dong and Yang, 2020) and DARTS (Liu et al., 2019) search spaces. We refer the reader to Appendix D for details about these spaces and Appendix F.4 for the experimental setup. We evaluate each method with 4 different random seeds.

Our contribution on these spaces is two-fold. Firstly, we study the effects of weight-entanglement on cell-based spaces in conjunction with single-stage methods. To this end, we entangle the weights of similar operations with different kernel sizes on both search spaces. For NB201, the weights of the 1×1 and 3×3 convolutions

| Search Type | Optimizer | Supernet | CIFAR-10 (%) | ImageNet (%) | Search Time (hrs) |
|---|---|---|---|---|---|
| Single-Stage | DrNAS | WS | 2.625 ± 0.075 | 26.290 | 9.1 |
| | **TangleNAS** | WE | **2.556 ± 0.034** | **25.691** | 7.4 |
| Two-Stage | SPOS+RS | WE | 2.965 ± 0.072 | 27.114 | 18.7 |
| | SPOS+ES | WE | 3.200 ± 0.065 | 27.320 | 14.8 |

Table 3: Comparison of test errors of single and two-stage methods on the DARTS search space.

are entangled, and in the DARTS search space, the weights of dilated and separable convolutions with kernel sizes 3×3 and 5×5 are (separately) entangled. Secondly, we study SPOS on the NB201 and DARTS search spaces. To the best of our knowledge, we are the first to study a two-stage method like SPOS in such cell search spaces.

Tables 3 and 4 show the results. For both search spaces, TangleNAS yields the best results, outperforming the single-stage baseline DrNAS with WS, as well as both SPOS variants. TangleNAS also significantly lowers the memory requirements and runtime compared to its weight-sharing counterpart. We note that overall, the SPOS methods are ineffective in these cell search spaces.

| Search Type | Optimizer | Supernet | CIFAR-10 | CIFAR-100 | ImageNet16-120 | Search Time (hrs) |
|---|---|---|---|---|---|---|
| Single-Stage | DrNAS | WS | **94.360 ± 0.000** | 72.245 ± 0.732 | **46.370 ± 0.000** | 20.9 |
| | **TangleNAS** | WE | **94.360 ± 0.000** | **73.510 ± 0.000** | **46.370 ± 0.000** | 20.4 |
| Two-Stage | SPOS+RS | WE | 89.107 ± 0.884 | 56.865 ± 2.597 | 31.665 ± 1.146 | 29.5 |
| | SPOS+ES | WE | 87.133 ± 2.605 | 56.463 ± 2.342 | 29.785 ± 3.015 | 26.7 |

Table 4: Comparison of test-accuracies of single and two-stage methods on NB201 search space.

### 4.3 Macro Search Spaces

Given the promising results of TangleNAS on the toy and cell-based spaces, we now extend our evaluation to the home base of two-stage methods. We study TangleNAS on a vision transformer space (AutoFormer-T and -S) and a convolutional space (MobileNetV3), both of which were previously proposed and examined using two-stage methods by Chen et al. (2021a) and Cai et al. (2020), respectively. Additionally, we explore a language model transformer search space centered around GPT-2 (Radford et al., 2019).

| Search Type | Optimizer | CIFAR-10 | | | | | CIFAR-100 | | | | |
|---|---|---|---|---|---|---|---|---|---|---|---|
| | | Inherit (%) | Fine-tune (%) | Retrain (%) | Params | FLOPS | Inherit (%) | Fine-tune (%) | Retrain (%) | Params | FLOPS |
| Single-Stage | TangleNAS | 93.570 | $97.702 \pm 0.017$ | $97.872 \pm 0.054$ | 8.685M | 1.946G | 76.590 | $82.615 \pm 0.064$ | $82.668 \pm 0.161$ | 8.649M | 1.939G |
| Two-Stage | SPOS+RS | **94.290** | $97.605 \pm 0.038$ | $97.767 \pm 0.024$ | 8.512M | 1.910G | **78.210** | $82.407 \pm 0.026$ | $82.210 \pm 0.142$ | 8.476M | 1.905G |
| | SPOS+ES | 94.100 | $97.632 \pm 0.047$ | $97.643 \pm 0.023$ | 7.230M | 1.659G | 77.970 | $82.517 \pm 0.140$ | $82.518 \pm 0.114$ | 8.245M | 1.859G |

Table 5: Test Accuracies on the AutoFormer-T space for CIFAR-10 and CIFAR-100 (across 4 random seeds)

**4.3.1 AutoFormer.** We evaluate TangleNAS on the AutoFormer-T and AutoFormer-S spaces introduced by Chen et al. (2021a), based on vision transformers. The search space consists of the choices of embedding dimensions and number of layers, and for each layer, its MLP expansion ratio and the number of heads it uses to compute attention. More details can be found in Table 20 in the appendix. The embedding dimension remains constant across the network, while the number of heads and the MLP expansion ratio change for each layer. This results in a search space of about $10^{13}$ architectures. We train our supernet using the same training hyperparameters and pipeline as used in AutoFormer, and use its evolutionary search as our baseline.

| SuperNet-Type | NAS Method | ImageNet (%) | Datasets | | | | | Params | FLOPS |
|---|---|---|---|---|---|---|---|---|---|
| | | | CIFAR-10 (%) | CIFAR-100 (%) | Flowers (%) | Pets(%) | Cars(%) | | |
| AutoFormer-T | SPOS+ES | 75.474 | 98.019 | 86.369 | **98.066** | 91.558 | 91.935 | 5.893M | 1.396G |
| AutoFormer-T | TangleNAS | **78.842** | 98.249 | **88.290** | **98.066** | **92.347** | **92.396** | 8.981M | 2.000G |
| AutoFormer-S | SPOS+ES | 81.700 | 99.100 | **90.459** | 97.900 | 94.853 | **92.545** | 22.900M | 5.100G |
| AutoFormer-S | TangleNAS | **81.964** | **99.120** | **90.459** | **98.326** | **95.070** | 92.371 | 28.806M | 6.019G |

Table 6: Evaluation on the AutoFormer-T space on downstream tasks. ImageNet-1k validation accuracies are obtained through inheritance, whereas the test accuracies for the other datasets are achieved through fine-tuning the ImageNet-pretrained model.

In Tables 5 and 6, we evaluate TangleNAS against AutoFormer on the AutoFormer-T and AutoFormer-S spaces. Interestingly, we observe that although AutoFormer sometimes outperforms TangleNAS upon inheritance from the supernet, the TangleNAS architectures are always better upon fine-tuning and much better upon retraining. For ImageNet-1k we obtain an improvement of 3.368% and 0.264% on AutoFormer-T and AutoFormer-S spaces, respectively (see Table 6).

**4.3.2 MobileNetV3.** Next, we study a convolutional search space based on the MobileNetV3 architecture. The search space is defined in Table 19 in the appendix and contains about $2 \times 10^{19}$ architectures. This follows from the search space designed by OFA (Cai et al., 2020), which searches for kernel size, number of blocks, and channel-expansion factor.

| Search Type | Optimizer | Top-1 acc (%) | Params | FLOPS |
|---|---|---|---|---|
| Single-Stage | TangleNAS | 77.424 | 7.580M | 528.80M |
| Two-Stage | OFA+RS | 77.046 | 6.870M | 369.160M |
| | OFA+ES | 77.050 | 7.210M | 420.500M |
| Largest Arch | - | 77.336 | 7.660M | 566.170M |

Table 7: Evaluation on MobileNetV3.

During the supernet training, we activate all choices in our supernet at all times. Table 7 shows that on this OFA search space, TangleNAS outperforms OFA (based on both unconstrained evolutionary and random search on the pre-trained OFA supernet). Notably, TangleNAS even yields an architecture with higher accuracy than the largest architecture in the space (while OFA yields worse architectures).

**4.3.3 Language Modeling (LM) Space**. Finally, given the growing interest in efficient large language models and recent developments in scaling laws (Kaplan et al., 2020; Hoffmann et al., 2022), we study our efficient and scalable TangleNAS method on a language-model space for two different scales. We create our language model space around a smaller version of nanoGPT model[1] and the model at its original size. In this transformer search space, we search (with 4 random seeds) for the embedding dimension, number of heads, number of layers, and MLP ratios (as defined in Table 18 of the appendix). We weight-entangle all of these by combi-superposition in four dimensions. For

| Architecture | Search-Type | Loss | Perplexity ↓ | Params | Inference time (s) |
|---|---|---|---|---|---|
| GPT-2 | Manual | 3.077 | 21.690 | 123.590M | 113.30 |
| **TangleNAS** | Automated | 2.904 | 18.243 | 116.519M | 102.50 |

(a) Comparison of fine-tuning architecture discovered by TangleNAS and GPT-2 on Shakespeare dataset.

| Search Type | Optimizer | Test loss | Perplexity ↓ | Params | Inference Time (s) |
|---|---|---|---|---|---|
| Single-Stage | TangleNAS | **1.412 ± 0.011** | **4.104** | 87.010M | 93.88 |
| Two-Stage | SPOS+RS | 1.433 ± 0.005 | 4.191 | 77.570M | 85.17 |
| | SPOS+ES | 1.444 ± 0.013 | 4.238 | 78.750M | 87.10 |

(b) Comparison of single and two-stage methods on language model search space. We report the test loss and perplexity on the TinyStories dataset.

Table 8: Evaluation on Language Model Space

each of these transformer search dimensions, we consider 3 different choices. We train our language models on the TinyStories (Eldan and Li, 2023) dataset (for the smaller version of nanoGPT) and on OpenWebText (Aaron et al., 2019) (for nanoGPT at its original size). Furthermore, we fine-tune the model trained on OpenWebText on the Shakespeare dataset. On the smaller scale, we beat the SPOS+ES and SPOS+RS baselines as presented in Table 8b. This improvement is statistically very significant with a two-tailed p-value of 0.0064 for ours v/s SPOS+ES and p-value of 0.0127 for ours v/s SPOS+RS. As shown in Table 8a, we obtain a smaller model on OpenWebText while achieving better perplexity after fine-tuning on Shakespeare[2]. This model is also very efficient during inference time in comparison to GPT-2.

## 5 Results and Discussion

For a more thorough evaluation, we now compare different properties of single and two-stage methods, focusing on their any-time performance, the impact of the train-validation split ratios, and the Centered Kernel Alignment (Kornblith et al., 2019) (see Appendix H) of the supernet feature maps. Additionally, we study the effect of pretraining, fine-tuning and retraining on the AutoFormer space for CIFAR-10 and CIFAR-100, as well as the downstream performance of the best model pre-trained on ImageNet across various classification datasets. We conclude by discussing the insights derived from TangleNAS in designing architectures on real world tasks.

**Space and time complexity**. In practice, we observe that vanilla gradient-based NAS methods are memory and compute expensive in comparison to both two-stage methods and our TangleNAS approach with weight-superposition. While the time and space complexity of single-stage methods is $\mathcal{O}(n)$, where $n$ is the number of operation choices, TangleNAS, similar to two-stage methods, maintains $\mathcal{O}(1)$. On the NB201 and DARTS search spaces we observe a 25.28% and 35.54% reduction in memory requirements for TangleNAS over DrNAS with WS. This issue only exacerbates for weight-entangled spaces like AutoFormer, MobileNetV3 and GPT, making the application of vanilla gradient-based methods practically infeasible.

**Anytime performance**. NAS practitioners often emphasize rapid discovery of competitive architectures. This is especially important given the rising costs of training large and complex neural networks, like Transformers. Thus, strong *anytime performance* is crucial for the practical deployment of NAS in resource-intensive environments. Therefore, we examine the anytime performance

---

[1] https://github.com/karpathy/nanoGPT
[2] https://huggingface.co/datasets/karpathy/tiny_shakespeare

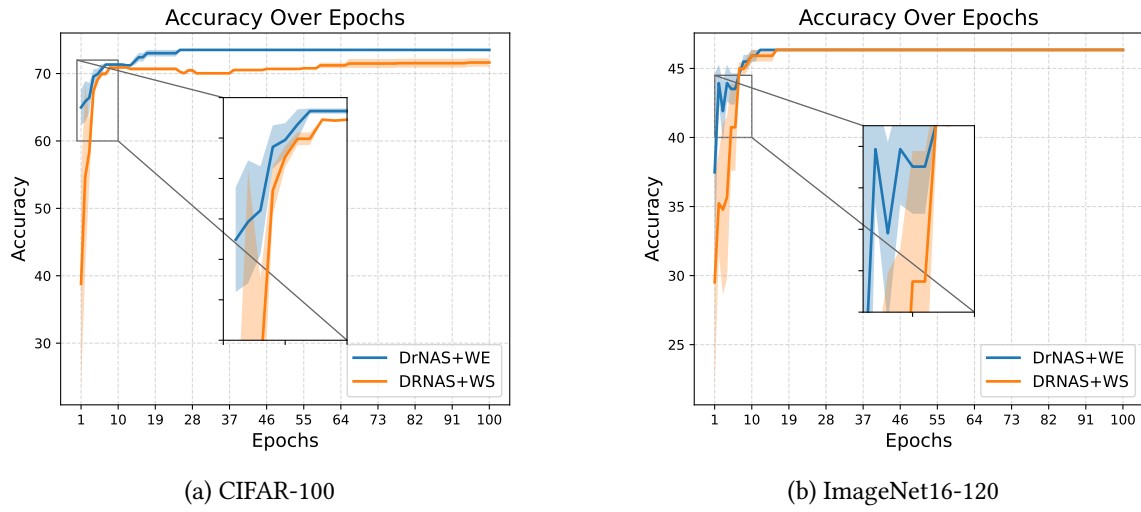

(a) CIFAR-100

(b) ImageNet16-120

Figure 4: Test accuracy evolution over epochs for NB201.

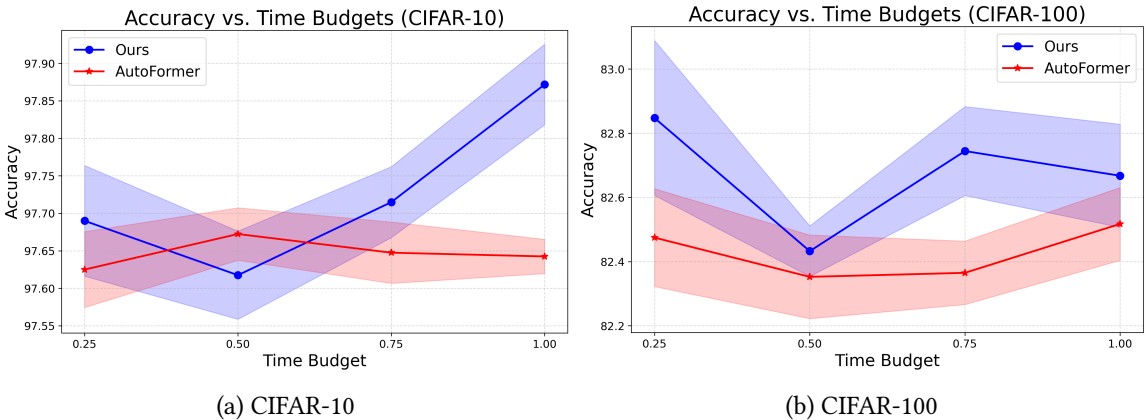

(a) CIFAR-10

(b) CIFAR-100

Figure 5: Any Time performance curves of AutoFormer vs. Ours.

of TangleNAS. Figure 4 demonstrates that TangleNAS (DrNAS+WE) is faster than its baseline method (DrNAS+WS). Similarly, Figure 5 shows that TangleNAS has better anytime performance than AutoFormer.

**Effect of fraction of training data**. One-shot NAS commonly employs a 50%-50% train-valid split for cell-based spaces and 80%-20% for weight-entangled spaces. To eliminate possible biases, we tested our method on various data splits within each search space. The findings, detailed in Section B, show consistent results across these splits. Specifically, single-stage methods prove to be robust and performant across different training fractions and search spaces, compared to two-stage methods.

### 5.1 Insights from NAS

**Architecture design insights**. In transformer spaces, reducing the *MLP-ratio* in the initial layers has a relatively low impact on performance (Figure 6) and can often work competitively or outperform handcrafted architectures. This observation is consistent across ViT and Language Model spaces. Conversely, *number of heads* and *embedding dimension* have a significant impact.

Pruning a few of the final layers also has a relatively low impact on performance. In the MobileNetV3 space, we find a strong preference for a larger *number of channels* and larger *network depth*. In contrast, we discover that scaling laws for transformers may not necessarily apply in

convolutional spaces, especially for *kernel sizes* - the earlier layers favor 5×5 kernel sizes while later ones prefer 7×7 (3 being the smallest and 7 the largest).

**Effect of pretraining, fine-tuning and re-training**. We examined the effects of inheriting, fine-tuning, and retraining in the AutoFormer space on the CIFAR-10 and CIFAR-100 datasets. We observe that retraining generally surpasses both fine-tuning and inheriting. This raises questions about the correlation between inherited and retrained accuracies of architectures in two-stage methods and the potential training interference identified by Xu et al. (2022). A strong correlation is crucial for two-stage methods, which use the performance of architectures with inherited

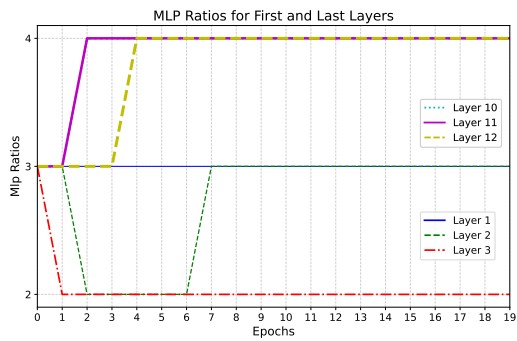

Figure 6: MLP ratio trajectory for LM. Number of layers range from 1-12 and MLP ratio choice can be 2, 3 or 4.

weights as a proxy for true performance in the black-box search. Indeed, while the SPOS+RS and SPOS+ES methods perform well with inherited weights, TangleNAS exceeds their performance after fine-tuning and retraining.

**ImageNet-1k pre-trained architecture on downstream tasks**. Lastly, we study the impact on fine-tuning the best model obtained from the search on downstream datasets. We follow the fine-tuning pipeline proposed in AutoFormer and fine-tune on different fine- and coarse-grained datasets. We observe from Table 6 that the architecture discovered by TangleNAS on ImageNet is much more performant in fine-tuning to various datasets (CIFAR-10, CIFAR-100, Flowers, Pets and Cars) than the architecture discovered by SPOS.

## 6 Conclusion and Broader Impact

In this paper, we compare single-stage and two-stage NAS methods, traditionally used for different search spaces, and introduce single-stage NAS to weight-entangled spaces, usually the domain of two-stage methods. We empirically evaluate our single-stage method, TangleNAS, on a diverse set of weight-entangled search spaces and tasks, showcasing its ability to outperform conventional two-stage NAS methods while enhancing search efficiency. Our positive results on macro-level search spaces suggest this approach could advance the development of modern architectures like Transformers within the NAS community. A recent work (Klein et al., 2024) starts training of the supernet from the largest pretrained model, subsequently fine-tuning it. Our method, which now renders single-stage methods applicable to broader families of search spaces (e.g., transformers), can similarly benefit from initialization with pretrained models, achieving additional computational savings. We leave this for future work.

This study addresses the high energy consumption associated with neural architecture search, particularly in black-box techniques that demand extensive computational resources to train many architectures from scratch. Our proposed method falls in the family of one-shot NAS methods, and hence significantly reduces energy usage and identifies efficient architectures with far less energy consumption than manual tuning.

## Acknowledgments

This research was partially supported by the following sources: TAILOR, a project funded by EU Horizon 2020 research and innovation programme under GA No 952215; the Deutsche Forschungsgemeinschaft (DFG, German Research Foundation) under grant number 417962828; the European Research Council (ERC) Consolidator Grant "Deep Learning 2.0" (grant no. 101045765). Robert Bosch GmbH is acknowledged for financial support. The authors acknowledge support from ELLIS and ELIZA. Funded by the European Union. Views and opinions expressed are however those of the author(s) only and do not necessarily reflect those of the European Union or the ERC. Neither the European Union nor the ERC can be held responsible for them.



.

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

## B  Training across data fractions

TangleNAS is more robust across dataset fractions for network weights and architecture optimization as seen from Tables 9, 10, 11, and 12.

| Search Type | Optimizer | Train portion | CIFAR-10 (%) | CIFAR-100 (%) |
|---|---|---|---|---|
| Single-Stage | TangleNAS | 50% | 97.715 ± 0.088 | 82.538 ± 0.118 |
| | | 80% | **97.872 ± 0.054** | **82.668 ± 0.161** |
| Two-Stage | SPOS+RS | 50% | 97.680 ± 0.026 | 82.537 ± 0.280 |
| | | 80% | 97.767 ± 0.024 | 82.210 ± 0.142 |
| | SPOS+ES | 50% | 97.77 ± 0.038 | 82.354 ± 0.120 |
| | | 80% | 97.642 ± 0.023 | 82.517 ± 0.114 |

Table 9: Evaluation on the AutoFormer-T space for CIFAR-10 and CIFAR-100.

| Search Type | Optimizer | Train portion | Supernet type | Test acc (%) |
|---|---|---|---|---|
| Single-Stage | DrNAS | 50% | WS | 10 |
| | | 80% | | 10 |
| | TangleNAS | 50% | WE | **83.020 ± 0.000** |
| | | 80% | | 82.495 ± .0.461 |
| Two-Stage | SPOS+RS | 50% | WE | 81.253 ± 0.672 |
| | | 80% | | 81.345 ± 0.383 |
| | SPOS+ES | 50% | WE | 81.890 ± 0.800 |
| | | 80% | | 82.322 ± 0.604 |
| Optimum | - | - | - | 84.41 |

Table 12: Evaluation on toy conv-macro search space on CIFAR-10 dataset.

| Search Type | Optimizer | Train portion | Supernet | Accuracy (%) | | |
|---|---|---|---|---|---|---|
| | | | | CIFAR-10 | CIFAR-100 | ImageNet16-120 |
| Single-Stage | DrNAS | 50% | WS | **94.36 ± 0.000** | 72.245 ± 0.732 | **46.37 ± 0.00** |
| | | 80% | | **94.36 ± 0.00** | 71.153 ± 0.697 | **46.37 ± 0.00** |
| | TangleNAS | 50% | WE | **94.36 ± 0.00** | **73.51 ± 0.000** | **46.37 ± 0.00** |
| | | 80% | | **94.36 ± 0.00** | **73.51 ± 0.000** | **46.37 ± 0.00** |
| Two-Stage | SPOS+RS | 50% | WE | 89.107 ± 0.884 | 56.865 ± 2.597 | 31.665 ± 1.146 |
| | | 80% | | 87.778 ± 2.446 | 53.68 ± 4.174 | 30.545 ± 3.643 |
| | SPOS+ES | 50% | WE | 87.133 ± 2.605 | 56.463 ± 2.342 | 29.785 ± 3.015 |
| | | 80% | | 89.095 ± 0.825 | 56.363 ± 4.724 | 30.935 ± 3.546 |

Table 10: Comparison of test accuracies of single and two-stage methods with WS and WE on NB201 search space.

| Search Type | Optimizer | Train portion | Supernet type | Test acc (%) |
|---|---|---|---|---|
| Single-Stage | DrNAS | 50% | WS | 91.19 ± 0.049 |
| | | 80% | | 91.125 ± 0.033 |
| | TangleNAS | 50% | WE | **91.3 ± 0.023** |
| | | 80% | | 91.065 ± 0.163 |
| Two-Stage | SPOS+RS | 50% | WE | 90.68 ± 0.253 |
| | | 80% | | 90.687 ± 0.110 |
| | SPOS+ES | 50% | WE | 90.318 ± 0.223 |
| | | 80% | | 90.595 ± 0.219 |
| Optimum | - | - | - | 91.63 |

Table 11: Evaluation on toy cell-based search space on Fashion-MNIST dataset.

| Search Type | Optimizer | Train portion | Supernet | CIFAR-10 (%) | ImageNet (%) |
|---|---|---|---|---|---|
| Single-Stage | TangleNAS | 50% | WE | **2.556 ± 0.034** | **25.69** |
| | | 80% | | 2.67 ± 0.076 | 25.742 |
| | DrNAS | 50% | WS | 2.625 ± 0.075 | 26.29 |
| | | 80% | | 2.580 ± 0.028 | **25.67** |
| Two-Stage | SPOS+RS | 50% | WE | 2.965 ± 0.072 | 27.114 |
| | | 80% | | 2.965 ± 0.072 | 27.114 |
| | SPOS+ES | 50% | WE | 3.200 ± 0.066 | 27.424 |
| | | 80% | | 3.002 ± 0.037 | 26.76 |

Table 13: Comparison of test errors of single and two-stage methods with WS and WE on DARTS search space.

## C Comparison with different gradient-based optimizers

We also compare DrNAS against different gradient-based NAS optimizers in Tables 15, 14, and 16 on the toy cell-based, toy macro and the AutoFormer search spaces. We observe that DrNAS outperforms GDAS and DARTS on all of these search spaces, showing its robust nature.

| Optimizer | Test-acc (%) |
|---|---|
| TangleNAS+DrNAS | **83.02** |
| SPOS+RS | 81.25 |
| SPOS+ES | 81.89 |
| TangleNAS+DARTS_v1 | 81.61 |
| TangleNAS+DARTS_v2 | 81.49 |
| TangleNAS+GDAS | 10 (degenerate) |

Table 14: Comparison of DrNAS with other gradient-based optimizers on the toy conv-macro search space on CIFAR-10.

| Optimizer | Test-acc (%) |
|---|---|
| TangleNAS+DrNAS | **90.930** |
| SPOS+RS | 90.688 |
| SPOS+ES | 90.595 |
| TangleNAS+DARTS_v1 | 89.905 |
| TangleNAS+DARTS_v2 | 90.747 |
| TangleNAS+GDAS | 90.618 |

Table 15: Comparison of DrNAS with other gradient-based optimizers on the toy cell-based search space on FashionMNIST dataset.

| Optimizer | CIFAR-10 | CIFAR-100 |
|---|---|---|
| TangleNAS+DrNAS | **97.872 ± 0.054** | **82.668 ± 0.161** |
| SPOS+RS | 97.767 ± 0.024 | 82.210 ± 0.142 |
| SPOS+ES | 82.518 ± 0.114 | 82.518 ± 0.114 |
| TangleNAS+DARTS_v1 | 97.672 ± 0.040 | 82.107 ± 0.392 |
| TangleNAS+GDAS | 97.45 ± 0.096 | 82.120 ± 0.281 |

Table 16: Comparison of DrNAS with other gradient-based optimizers on the AutoFormer-T space for CIFAR-10 and CIFAR-100.

## D  Search Space details

**Toy cell space.** This particular search space takes its inspiration from the prominently used Differentiable Architecture Search (DARTS) space, and is composed of diminutive triangular cells, with each edge offering four choices of operations: (a) Separable 3×3 Convolution, (b) Separable 5×5 Convolution, (c) Dilated 3×3 Convolution, and (d) Dilated 5×5 Convolution. The macro-architecture of the model comprises three cells of the types reduction, normal, and reduction again stacked together. Notably, we entangle the 3×3 and 5×5 kernel weights for each operation type, i.e., separable convolutions and dilated convolutions. We evaluate these search spaces and their architectures on the Fashion-MNIST dataset by creating a small benchmark, which we release here.

**Toy conv-macro space.** This toy space draws its inspiration from MobileNet-like spaces where we search for the number of channels and the kernel size of convolutional layers in a network (also referred to as a macro search space) for four convolutional layers. Every convolutional layer has a choice of three kernel sizes and number of channels. See Table 17 for more details. We evaluate this search space on the CIFAR-10 dataset by creating a small benchmark, which we release here.

| Architectural Parameter | Choices |
|---|---|
| Kernel sizes (all layers) | 3, 5, 7 |
| Number of channels (layer 1) | 8, 16, 32 |
| Number of channels (layer 2) | 16, 32, 64 |
| Number of channels (layer 3) | 32, 64, 128 |
| Number of channels (layer 4) | 64, 128, 256 |

Table 17: Toy Convolutional-Macro Search Space.

**NAS-Bench-201 Space.** The NAS-Bench-201 search space (Dong and Yang, 2020) has a single cell type which is stacked 15 times, with residual blocks (He et al., 2016) after the fifth and tenth cells. Each cell has 4 nodes, with each node connected to the previous ones by operations chosen from skip connection, 3×3 convolution, 1×1 convolution, 3×3 average-pooling, and *zeroize*, which zeros out the input feature map. There is limited scope for weight-entanglement in this space, given that only two of the five candidate operations contain learnable parameters. In this case, we entangle the 3×3 and 1×1 convolution kernels on every edge for every cell, thus obtaining parameter savings over traditional weight sharing.

**DARTS Search Space.** The DARTS (Liu et al., 2019) search space has two kinds of cells - *normal* and *reduction*. Reduction cells double the number of channels in its outputwhile halving the width and height of the feature maps. The supernet consists of 8 cells stacked sequentially, while the discretized network stacks 20 cells. Reduction cells are placed at 1/3 and 2/3 of the total depth of the network. Each cell takes two inputs - one each from the outputs of the two previous cells. There are 14 edges in each cell, with each edge in the supernet consisting of 8 candidate operations as follows: 3×3 max pooling, 3×3 average pooling, skip connect, 3×3 separable convolutions, 5×5 separable convolutions, 3×3 dilated convolutions, 5×5 dilated convolutions, and *none* (no operation).

| Architectural Parameter | Choices |
|---|---|
| Embedding Dimension | 384, 576, 768 |
| Number of Heads | 6, 8, 12 |
| MLP Expansion Ratio | 2, 3, 4 |
| Number of Layers | 5,6,7 |

Table 18: Choices for Language Model Space.

| Architectural Parameter | Choices |
|---|---|
| Kernel sizes (all layers) | 3, 5, 7 |
| Channel expansion (all layers) | 3,4,6 |
| Number of blocks | 2,3,4 |

Table 19: MobileNetV3 Search Space.

**AutoFormer Space and Language Model Space**. We present the details of the AutoFormer Space and the Language Model space is Tables 20 and 18, respectively.

## E  Methodological details

**Combi-Superposition**. Traditional cell-based search spaces primarily consider independent operations (e.g., convolution or skip). One-shot differentiable optimizers thus have their mixture operations tailored to these search spaces, which are not general enough to be applied to macro-level architectural parameters. Consider, e.g., the task of searching for the embedding dimension and the expansion ratio for a transformer. Here, a single operation, i.e., a linear expansion layer, has two different architectural parameters – one corresponding to the choice of embedding dimension and the other to the expansion ratio. To adapt single-stage methods to these *combined* operation choices in the search space, we propose the *combi-superposition operation*.

The combi-superposition operation simply takes the cross product of architectural parameters for the embedding dimension and expansion ratio and assigns its elements to *every combination* of these dimensions. This allows us to optimize jointly in this space without the need for a separate forward pass for each combination. Every combination maps to a unique sub-matrix of the operator weight matrix, indexed by both the embedding dimension and the expansion ratio. To address shape mismatches of the different operation weights during forward passes, every sub-matrix is zero-padded to match the shape of the largest matrix. See Algorithm 2 for more details, and Figure 3 for an overview of the idea.

| Architectural Parameter | Choices |
|---|---|
| Embedding dimension | 192, 216, 240 |
| Number of layers | 12, 13, 14 |
| MLP ratio (per layer) | 3.5, 4 |
| Number of heads (per layer) | 3, 4 |

Table 20: Choices for AutoFormer-T Search Space.

---

**Algorithm 2** Combi-Superposition Operation

---

*For ease of presentation, we show the algorithm for superimposing along two dimensions. However, in practice, we can super-impose an arbitrary number of dimensions (e.g., four dimensions in our experiments with NanoGPT).*

$embed\_dim \leftarrow [e_1, e_2, e_3, \ldots, e_n]$ {Choices for embedding dimension}
$expansion\_ratio \leftarrow [r_1, r_2, r_3, \ldots, r_m]$ {Choices for expansion ratio}
$\alpha \leftarrow [\alpha_1, \alpha_2, \alpha_3, \ldots, \alpha_n]$ {Architecture parameters for embedding dimension}
$\beta \leftarrow [\beta_1, \beta_2, \beta_3, \ldots, \beta_m]$ {Architecture parameters for expansion ratio}
$X \leftarrow input\_feature$
$W, b \leftarrow fc\_layer\_weight, fc\_layer\_bias$
$W_{mix} \leftarrow \mathbf{0}$
$b_{mix} \leftarrow \mathbf{0}$
**for** $i \leftarrow 1$ to $n$ **do**
    **for** $j \leftarrow 1$ to $m$ **do**
        $W\_ij = W[: (embed\_dim[i] \times expansion\_ratio[j]), : embed\_dim[i]]$
        $b\_ij = b[: embed\_dim[i] \times expansion\_ratio[j]]$
        $W_{mix} \leftarrow W_{mix} + \text{normalize}(\alpha[i]) \cdot \text{normalize}(\beta[j]) \times \text{PAD}(W_{ij})$
        $b_{mix} \leftarrow b_{mix} + \text{normalize}(\alpha[i]) \cdot \text{normalize}(\beta[j]) \times \text{PAD}(b_{ij})$
    **end for**
**end for**
$Y \leftarrow X \cdot W_{mix} + b_{mix}$ {Compute the output of the FC layer with a mixture of weights and bias}
**return** $Y$

---

For completeness and to facilitate comparison with TangleNAS (Algorithm 1), we present Algorithm 3, which describes a generic two-stage method on a macro search space with weight entanglement (WE). Additionally, vanilla single-stage methods on cell-based weight sharing (WS) spaces follow Algorithm 4.

---

**Algorithm 3** Weight Entanglement (Two-Stage)

---

1: **Input**: $M \leftarrow$ number of cells, $N \leftarrow$ number of operations
    $\mathcal{O} \leftarrow [o_1, o_2, o_3, \ldots o_N]$
    $\mathcal{W}_{max} \leftarrow \cup_{i-1}^{N} w_i$
    $\eta = $ learning rate of $\mathcal{W}_{max_\mathcal{O}}$
2: $Cell_j \leftarrow DAG(\mathcal{O}_|, \mathcal{W}_{max_|})$ /* defined for j=1...M */
3: $Supernet \leftarrow \cup_i^M Cell_i$
4: /* example of forward propagation on the cell */
5: **for** $j \leftarrow 1$ to $M$ **do**
6:     $i \sim \mathcal{U}(1, N)$
7:     /* Slice weight matrix corresponding to operation */
8:     $o_j(x, \mathcal{W}_{max}) = o_{(j,i)}(x, \mathcal{W}_{max}[:i])$
9: **end for**
10: /* weights update */
11: $\mathcal{W}_{max}[:i] = \mathcal{W}_{max}[:i] - \eta \nabla_{\mathcal{W}_{max}}[:i] \mathcal{L}_{train}(\mathcal{W}_{max})$
12: /* Search */
13: $Supernet^* \leftarrow$ pre-trained supernet
14: $selected\_arch \leftarrow$ Evolutionary-Search$(Supernet^*)$

---

**Algorithm 4** Weight Sharing (Single-Stage)

---

1: **Input**: $M \leftarrow$ number of cells, $N \leftarrow$ number of operations
    $\mathcal{O} \leftarrow [o_1, o_2, o_3, \ldots o_N]$
    $\mathcal{W}_\mathcal{O} \leftarrow [w_1, w_2, w_3, \ldots w_N]$
    $\mathcal{A} \leftarrow [\alpha_1, \alpha_2, \alpha_3, \ldots \alpha_N]$
    $\gamma = $ learning rate of $\mathcal{A}$
    $\eta = $ learning rate of $\mathcal{W}_\mathcal{O}$
    $f$ is a function or distribution s.t. $\sum_{i=1}^N f(\alpha_i) = 1$
2: $Cell_i \leftarrow DAG(\mathcal{O}_|, \mathcal{W}_{\mathcal{O}_|})$ /* defined for i=1...M */
3: $Supernet \leftarrow \cup_i^M Cell_i \cup \mathcal{A}$
4: /* example of forward propagation on the cell */
5: **for** $j \leftarrow 1$ to $M$ **do**
6:     /* Compute mixture operation as weighted sum of output of operations*/
7:     $\overline{o_j(x, \mathcal{W}_\mathcal{O})} = \sum_{i=1}^N f(\alpha_i) o_{(j,i)}(x, w_{(j,i)})$
8: **end for**
9: /* weights and architecture update */
10: $\mathcal{A} = \mathcal{A} - \gamma \nabla_\mathcal{A} \mathcal{L}_{val}(\mathcal{W}_\mathcal{O}^*, \mathcal{A})$
11: $\mathcal{W}_\mathcal{O} = \mathcal{W}_\mathcal{O} - \eta \nabla_{\mathcal{W}_\mathcal{O}} \mathcal{L}_{train}(\mathcal{W}_\mathcal{O}, \mathcal{A})$
12: /* Search */
13: $selected\_arch \leftarrow \arg\max(\mathcal{A})$

---

**Compatibility issues between weight-entanglement and gradient-based methods**. We address the incompatibility between gradient-based NAS methods and weight-entangled (WE) spaces as follows (where $n$ refers to the number of operation choices):

- Gradient-based NAS methods do not share or entangle weights among competing operations, which increases their GPU memory footprint. We tackle this issue by adopting weight-entanglement from two-stage methods, thereby reducing the parameter size of the supernet from $\mathcal{O}(n)$ to $\mathcal{O}(1)$.

- Gradient-based NAS methods compute a weighted combination of output of the operations. Even after entangling the operation weights, this approach leads to increased GPU memory usage because all intermediate competing activations/features need to be preserved in memory. Additionally, this method

does not scale well with an increase in the number of operation choices, as GPU memory consumption during forward propagation scales as $\mathcal{O}(n)$. To address this, we propose weight superposition, which computes the weighted combination directly within the space of entangled weights.

- In vanilla gradient-based methods, a forward pass is computed independently for each operation choice, resulting in a time and memory complexity of $\mathcal{O}(n)$. In contrast, TangleNAS computes only a single forward pass using the superimposed weights, reducing the cost of the forward pass to $\mathcal{O}(1)$.

**Details on Figure 1.** In the overview Figure 1, we use rounded squares to represent the input and output feature maps of a convolution. The colored cubes represent the convolutional kernels, while the colored rectangles denotes non-convolutional architectural choices. These two differ primarily in how their weights are entangled.

Consider Figure 1 (a), which illustrates the forward pass of an input feature map through the candidate operations on an edge of the supernet in a two-stage weight-entanglement method (such as OFA). In two-stage methods, random paths through the supernet are sampled and trained in the first stage. In this figure, thick lines indicate the paths sampled in a given step. The weights of the operation choices, depicted in the figure by colored cubes and rectangles, overlap with one another, showing that all the operations use slices of a common, large weight matrix. We show three choices of convolutions, each of a different kernel size (say, 1×1, 3×3, and 5×5), represented by cubes of varying sizes. The 1×1 and 3×3 convolutions use slices of the larger 5×5 convolution as their weights, represented in the figure by nesting the smaller kernels inside the larger ones. Since only one operation is sampled at a time (in this case, the orange kernel), there is only one output feature map. This feature map, after global average pooling, is then passed through one of the three non-convolutional operation choices. The operation choices here may be the embedding dimension, for example, and different slices of the largest feedforward network are used for the choices of embedding dimensions. At the end of the first stage in Figure 1 (a), the supernet has been trained along different paths. In the second stage, paths are sampled from the trained supernet using black-box methods (such as random or evolutionary search) and evaluated on the dataset to obtain the *optimal architecture*, which we represent with *optim arch*.

Now, consider Figure 1 (b), which represents the forward pass in a single-stage weight-sharing method (such as DARTS). As you can see, there is no nesting of the weights of the three convolutions or feedforward networks in this case, indicating that each has its own distinct set of weights. Naturally, this will incur more GPU memory usage, as more weights need to be stored and their gradients computed. The outputs from each of these convolutional (or feedforward) operations are then weighted by $\alpha_1$, $\alpha_2$, and $\alpha_3$ (or $\beta_1$, $\beta_2$, and $\beta_3$), which represent the architectural parameters of the operations. These weighted feature maps are then summed up to produce the output feature maps.

The optimization loop of single-stage weight-sharing methods, shown in Figure 1(b), aims to find the optimal values for the architectural parameters $\alpha$ and $\beta$. We obtain the *discretized* model by selecting only the operations in the supernet with the highest values of these parameters at the end of this loop. Specifically, the convolutional kernel and non-convolutional operator (in this case, the feedforward network) with the maximum values of $\alpha$ and $\beta$ are represented as argmax($\alpha_1$, $\alpha_2$, $\alpha_3$) and argmax($\beta_1$, $\beta_2$, $\beta_3$), respectively. The resulting discretized model, or optimal architecture, is denoted as *optimal arch*. Note that these methods do not require black-box search, as the optimal architecture can be obtained directly from the learned architectural parameters.

In Figure 1 (c), we depict our hybrid framework, which utilizes both entangled (nested) weights and architectural parameters. This approach reduces GPU memory usage due to weight entanglement and allows for the optimal architecture to be obtained directly from the learned parameters, eliminating the need for a black-box search.

## F Experimental Setup

### F.1 Toy Search Spaces

Below are the hyperparameter settings for the two toy spaces for *TangleNAS* and SPOS. All experiments were run on a single RTX-2080 GPU.

| Search Space | Cell Space | Conv Macro |
|---|---|---|
| Epochs | 100 | 100 |
| Learning rate (LR) | 0.1 | 3e-4 |
| Min. LR | 0.001 | 0.0001 |
| Optimizer | SGD | Adam |
| Architecture LR | 0.0003 | 0.0003 |
| Batch Size | 64 | 64 |
| Momentum | 0.9 | 0.9 |
| Nesterov | True | False |
| Weight Decay | 0.0005 | 0.0005 |
| Arch Weight Decay | 0.001 | 0.001 |
| Regularization Type | L2 | L2 |
| Regularization Scale | 0.001 | 0.001 |

Table 21: Configurations used in the DrNAS experiments on Toy Spaces.

| Search Space | Cell Space | Conv Macro |
|---|---|---|
| Epochs | 250 | 250 |
| Learning rate (LR) | 0.1 | 3e-4 |
| Min. LR | 0.001 | 0.0001 |
| Optimizer | SGD | Adam |
| Batch Size | 64 | 64 |
| Momentum | 0.9 | 0.9 |
| Nesterov | True | False |
| Weight Decay | 0.0005 | 0.0005 |

Table 22: Configurations used in the SPOS experiments on Toy Spaces.

## F.2 Language Model

We use the AdamW optimizer in all experiments related to language modeling. Other hyperparameter choices are as specified in Table 23. Below are the hyperparameter settings for *TangleNAS*. We run experiments on TinyStories on 8 RTX-2080 GPUs and experiments on OpenWebText on 8 A6000 GPUs.

| Search Space | Small-LM |
|---|---|
| Learning rate (LR) | 5e-4 |
| Min LR | 5e-5 |
| Beta2 | 0.99 |
| Warmup Iters | 100 |
| Max Iters | 6000 |
| Lr decay iters | 6000 |
| Batch size | 12 |
| Weight decay | 1e-1 |

Table 23: Configurations used in DrNAS on the Language Model Spaces.

### F.3 AutoFormer and OFA

We use a 50%-50% train and validation split for the CIFAR-10 and CIFAR-100 datasets for the cell-based spaces and a 80%-20% for the weight-entangled spaces. We use the official source code of AutoFormer available at code autoformer for all the AutoFormer experiments on CIFAR-10 and CIFAR-100, closely following the AutoFormer training pipeline and search space design. AutoFormer explored three transformer sizes: Autoformer-T (tiny), AutoFormer-S (small), and AutoFormer-B (base). We restrict ourselves to Autoformer-T.

For baselines like OFA and AutoFormer, we follow their respective recipes to obtain the train-validation split for ImageNet-1k. Our models were trained on 2xA100s with the same effective batch-size as AutoFormer. For MobileNetV3 from Once-for-All, we use the same training hyperparameters as the baseline found here (in addition to architectural parameters same as Table 24). We run experiments on CIFAR-10, CIFAR-100, and ImageNet1-k on 4 A100 GPUs.

#### F.3.1 AutoFormer Fine-tuning.

**CIFAR-10 and CIFAR-100 pretrained supernet.** We fine-tune the CIFAR-10 and CIFAR-100 selected networks (after inheriting them from the supernet) for 500 epochs. We set the learning rate to 1e-3, the warmup epochs to 5, the warmup learning rate to 1e-6, and the minimum learning rate to 1e-5. All other hyperparameters are set the same as in Appendix F.3.

**ImageNet pretrained supernet.** We follow the DeiT (Touvron et al., 2021) fine-tuning pipeline as used in AutoFormer, to finetune on downstream tasks. Specifically, we set the epochs to 1000, the warmup epochs to 5, the scheduler to cosine, the mixup to 0.8, the smoothing to 0.1, the weight decay to 1e-4, the batch size to 64, the optimizer to SGD, the learning rate to 0.01 and the warmup learning rate to 0.0001 for all datasets. Fine-tuning is performed on 8 RTX-2080 GPUs.

### F.4 NB201 and DARTS

For single-stage optimizers, the supernet was trained with four different seeds. The supernet with the best validation performance among these four was discretized to obtain the final model, which was then trained from scratch four times to obtain the results shown in the table. For two-stage methods, we again train the supernet four times and perform Random Search (RS) and Evolutionary Search (ES) on each one. The best model obtained across all four supernets for both methods was then trained from scratch with four seeds to compute the final results. For DrNAS, we follow the same procedure as suggested by the authors across all search spaces. To accommodate multiple training recipes, we have developed a configurable training pipeline. The configurations for DrNAS and SPOS are shown in Tables 24 and 25, respectively. We run our search on a single RTX-2080 for both DARTS and NB201, and train the DARTS architectures from scratch on 8 RTX-2080 GPUs.

| Search Space | DARTS | NB201 |
|---|---|---|
| Epochs | 50 | 100 |
| Learning rate (LR) | 0.1 | 0.025 |
| Min. LR | 0.0 | 0.001 |
| Architecture LR | 0.0006 | 0.0003 |
| Batch Size | 64 | 64 |
| Momentum | 0.9 | 0.9 |
| Nesterov | True | False |
| Weight Decay | 0.0003 | 0.0003 |
| Arch Weight Decay | 0.001 | 0.001 |
| Partial Connection Factor | 6 | - |
| Regularization Type | L2 | L2 |
| Regularization Scale | 0.001 | 0.001 |

Table 24: Configurations used in the DrNAS experiments.

| Search Space | DARTS | NB201 |
|---|---|---|
| Epochs | 250 | 250 |
| Learning rate (LR) | 0.025 | 0.025 |
| Min. LR | 0.001 | 0.001 |
| Architecture LR | 0.0003 | 0.0003 |
| Batch Size | 256 | 64 |
| Momentum | 0.9 | 0.9 |
| Nesterov | True | True |
| Weight Decay | 0.0005 | 0.0005 |
| Arch Weight Decay | 0.001 | 0.001 |

Table 25: Configurations used in the SPOS experiments.

# G  Optimal architectures derived

## G.1  DARTS

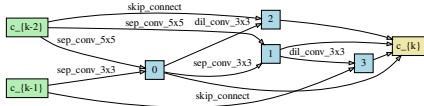

Figure 7: DRNAS with WE normal cell

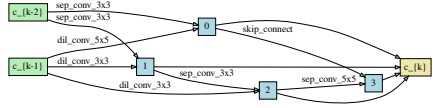

Figure 8: DRNAS with WE reduction cell

## G.2  Small LM

num_layers: 7, embed_dim: 768, num_heads: [12, 12, 12, 12, 12, 12, 12], mlp_ratio: [3, 4, 4, 4, 4, 4, 4]

## G.3  AutoFormer

### G.3.1  CIFAR-10.

**50%50% split.** mlp_ratio:[4, 4, 4, 4, 4, 4, 4, 4, 4, 3.5, 4, 4, 3.5, 4], num_heads:[4, 4, 4, 4, 4, 4, 4, 4, 4, 4, 4, 4, 4, 4], num_layers: 14 embed_dim: 216

**80%-20% split.** mlp_ratio:[4, 4, 4, 4, 4, 4, 4, 4, 3.5, 4, 4, 4, 3.5, 3.5], num_heads:[4, 4, 4, 4, 4, 4, 4, 4, 4, 4, 4, 4, 4, 4], depth: 14, embed_dim: 240

### G.3.2  CIFAR-100.

**50%-50% split.** mlp_ratio: [4,4,4,4,4,4,4,4,3.5,4,4,4,4,4], num_heads:[4,4,4,4,4,4,4,4,4,4,4,4,4,4], depth: 14, embed_dim: 216

**80%-20% split.** mlp_ratio:[3.5,4,4,4,4,4,4,4,4,4,4,4,4,4], num_heads:[4,4,4,4,4,4,4,4,4,4,4,4,4,4, depth: 14, embed_dim: 240

### G.3.3 ImageNet1-k.
mlp_ratio:[4,4,4,4,4,4,4,4,4,4,4,4,4,4], num_heads:[4,4,4,4,4,4,4,4,4,4,4,4,4,4, depth: 14, embed_dim: 240

## G.4 MobileNetV3

Kernel_sizes:[7,5,5,7,5,5,7,7,5,7,7,7,5,7,7,5,5,7,7,5],Channel_expansion_factor:[6,6,6,6,6,6,6,6,6,6,6,6,6,6,6,6,6,6,6,6], Depths : [4, 4, 4, 4, 4]

## G.5 Toy Spaces

### G.5.1 Toy cell (our best architecture) .

**50%-50% split.** Genotype(normal=[('dil_conv_3x3', 0), ('dil_conv_3x3', 0), ('sep_conv_3x3', 1)], normal_concat=range(1, 3), reduce=[('sep_conv_3x3', 0), ('sep_conv_3x3', 0), ('dil_conv_3x3', 1)], reduce_concat=range(1, 3))

**80%-20% split.** Genotype(normal=[('dil_conv_3x3', 0), ('dil_conv_5x5', 0), ('sep_conv_3x3', 1)], normal_concat=range(1, 3), reduce=[('sep_conv_3x3', 0), ('sep_conv_5x5', 0), ('dil_conv_3x3', 1)], reduce_concat=range(1, 3))

### G.6 Toy conv-macro (our best architecture)

**50%-50% split:.** Channels = [32, 64, 128, 64], Kernel Sizes = [5, 5, 7, 7].

**Train-Val fraction 80%-20%:.** Channels = [32, 64, 128, 64], Kernel Sizes = [5, 5, 7, 7].

## H Architecture Representation Analysis

**CKA.** Centered Kernel Alignment (CKA) (Kornblith et al., 2019) is a metric based on the Hilbert-Schmidt Independence Criterion (HSIC). It is designed to model the similarity between representations in neural networks. In this section, we analyze the CKA between structurally identical layers in the inherited, fine-tuned, and retrained networks in the AutoFormer-T space. Specifically, the aim is to assess how similar the inherited and fine-tuned representations are to those of the models trained from scratch. Table 26 presents the average CKA values for a fixed subset of the CIFAR-10 and CIFAR-100 datasets. We find that the representations learned by the single-stage supernet are more similar to those of the architectures that are fine-tuned or trained from scratch. This observation underscores potential issues with using inherited weights from the supernet as a proxy for search in two-stage methods, as noted by Xu et al. (2022).

| Model | Inherit v/s Retrain | | Fine-Tune v/s Retrain | |
|---|---|---|---|---|
| | CIFAR-10 | CIFAR-100 | CIFAR-10 | CIFAR-100 |
| TangleNAS | **0.4630** | **0.5853** | 0.5712 | **0.6527** |
| SPOS+ES | 0.4581 | 0.5793 | 0.5694 | 0.6309 |
| SPOS+RS | 0.4412 | 0.583 | **0.5797** | 0.6389 |

Table 26: CKA correlation between layers.

## I Limitations and Future Work

Currently, TangleNAS is designed to optimize a single objective, such as a chosen performance metric. However, in practice, there may be multiple objectives of interest, including hardware efficiency, robustness, and fairness (Dooley et al., 2023). Additionally, it would be valuable to explore and apply our findings across various applications in computer vision and natural language processing.

