# OpenReview forum: "Weight-Entanglement Meets Gradient-Based Neural Architecture Search"
_automl.cc/AutoML/2024/Conference — AutoML 2024_

### Official Review · Reviewer_77fw · 2024-03-25

**Potential Impact On The Field Of Automl Rating:** 2
**Technical Quality And Correctness Rating:** 3
**Clarity Rating:** 2
**Actions Required To Increase Overall Recommendation:** Please refer to the previous comments.

**Summary Of Contributions:**

Weight entanglement is not directly compatible with gradient-based NAS methods, so the authors would like to bridge these two parts. The authors claimed their findings reveal that this integration of weight-entanglement and gradient-based NAS brings forth the various benefits of gradient-based methods while preserving the memory efficiency of weight-entangled spaces.

**Clarity:**

The paper is not clear, even though I went through it several times, I still have a lot of doubts. If authors present something, they are required to explain everything clearly, or it is even better if they remove the unexplained parts, like Figure 1 (b), and (c).

1) In Figure 1, what do the cubics and rectangles with different colors mean? Similarly, please elaborate on the annotation of "f fn_dim = argmax...", "optimal arc", "\beta_1, \beta_2"...

2) From lines 44 to 60, the authors try to explain the distinguishing features of the NAS paradigm, but it is ambiguous due to the lack of explanation of (b) and (c).

3) In the Related Work, authors tried to introduce "Weight-sharing" and "Weight-entanglement". However they only mentioned WE as "provides a more effective way of weight-sharing exclusive to macro-level architectural spaces" and then introduced many papers using WE. So WE is not introduced clearly at all.

4) In Section 3, the authors mentioned "Weight-Superposition" but didn't explain it; they just categorized the "Weight-Superposition" in the caption in Figure 2. At least, the authors should introduce the motivation of the "Weight superposition"  and explain it further.

5) The main motivation of the paper is that weight-entanglement is not directly compatible with gradient-based NAS methods, and then the authors would like to bridge these two parts. However, I failed to understand why the two domains are not compatible and how the authors are able to solve the incompatibility. This is the key point of the whole paper, but the authors didn't emphasize it or show it in a clear way.

**Overall Review:**

I couldn't ignore the improved performance of the paper. But the presentation of the paper should be improved, as I said in the "Clarity."

Meanwhile, I didn't see many novelties in this paper. The paper proposed an algorithm incorporating gradient methods and weight entanglement techniques for neural architecture search problems. For both gradient methods and weight entanglement techniques, the authors just use the existing ones without many novelties. The paper probably is cited only in the NAS domain and might have a huge impact on the whole AutoML domain due to their hybrids of existing techniques.

**Potential Impact On The Field Of Automl:**

The paper proposed an algorithm incorporating gradient methods and weight entanglement techniques for neural architecture search problems. For both gradient methods and weight entanglement techniques, the authors just use the existing ones without many novelties. The paper is probably cited only in the NAS domain and might have a huge impact on the whole AutoML domain because they only hybridize the existing methods of gradient methods and weight-entanglement techniques.

**Reproducibility:**

The code has been uploaded, but I didn't check it in detail.

**Review Confidence:**

3

**Review Rating:**

7

**Review Summary:**

Please refer to the "Overall Review".

**Technical Quality And Correctness:**

According to the experimental results, the performance of the proposed TangleNAS seems to be good. Because the authors conducted their experiments on different benchmark problems,

However, I still have some concerns about the "Any Time performance curves", from Figure 5, why does the accuracy decrease a little bit given a longer timeout?

For NAS, in general, there are two general objectives: find a new architecture uncovered before; and get a good enough structure within a timeout and limited resources. From the experiments in the paper, all results try to achieve the second objective. How about the algorithm's achievement in the discovery of new architecture [1]?

[1] White, Colin, et al. "Neural architecture search: Insights from 1000 papers." arXiv preprint arXiv:2301.08727 (2023).

---

### Official Review · Reviewer_EcSa · 2024-03-26

**Potential Impact On The Field Of Automl Rating:** 3
**Technical Quality And Correctness Rating:** 4
**Clarity:** The paper is written clearly.
**Clarity Rating:** 4
**Actions Required To Increase Overall Recommendation:** Please refer to "overall review".

**Summary Of Contributions:**

This paper extends weight sharing common in NAS with weight entanglement to allow more intricate weight sharing. Specifically, the authors propose two major modifications to weight-sharing by 1) sharing the weights on every edge with the largest operator rather than the mixed ops to reduce the supernet size and 2) propose weight-superposition by zero-padding the operations to the largest operation. The authors then empirically show their method across a comprehensive set of search spaces to show their effectiveness over previous weight-sharing-based NAS approaches.

**Overall Review:**

Overall, I think the paper presents a reasonable and straightforward modification to the weight-sharing NAS setup. Extensive experimental validation was also performed across many search spaces, ranging from the classical CNN search spaces in early NAS papers to emerging paradigms such as Autoformer and language modelling. While the idea may be viewed a straightforward adaptation of ideas from another community to NAS, I do think the work is solid and merit acceptance. My only concern is that while comprehensive and I commend the authors for their effort, the extent of gain seems marginal across many search spaces, possibly also due to the fact that some of the search spaces are already rather saturated, but I'd appreciate if the authors could provide some comment on this.

**Potential Impact On The Field Of Automl:**

The paper is likely to spur some interest in the NAS community, but the paper features a rather minor modification to the overall NAS pipeline and the extent of the gain not always significant over all search spaces. I think the idea proposed is straightforward, and will likely spur a moderate amount of interest.

**Review Confidence:**

3

**Review Rating:**

7

**Review Summary:**

The paper is well-written and presents solid research ideas that are of interest to at least some members of the AutoML community. The experiments are very thorough, and I recommend acceptance of this paper.

**Technical Quality And Correctness:**

The method is sound and comprehensive experimental evaluation is performed to validate the method across a broad range of search spaces, and properly used error bars and standard deviations to run the experiments for multiple rounds for consistency; I do not see any major soundness issue in the paper.

---

### Official Review · Reviewer_3qdv · 2024-03-27

**Potential Impact On The Field Of Automl Rating:** 2
**Technical Quality And Correctness Rating:** 4
**Clarity:** The paper is clearly written.
**Clarity Rating:** 4
**Actions Required To Increase Overall Recommendation:** 1. A new search strategy rather than …

**Summary Of Contributions:**

Conventionally, neural architecture search over macro search spaces is conducted using a Blackbox optimization algorithm such as evolutionary algorithm combined with weight entanglement. This is conducted in two stages. On the other hand, NAS on micro search spaces is conducted with weight sharing approach over the super-net and using a gradient based optimization algorithm. This is conducted in a single stage. The authors present an approach to combine the gradient-based weight sharing approach with the weight-entanglement approach. The authors propose their algorithm as Tangle-NAS and have conducted comprehensive survey to present its credibility.

**Details Of Ethical Concerns:**

None.

**Overall Review:**

The authors address a topic in traditional NAS which has not been covered upon for a long time, i.e. the combination of both micro and macro search spaces with a suitable strategy. While the area is niche, the approach taken by authors is obvious. The paper is well-written except for the formulation section. More details in the Formulation section can benefit the readers.

I also believe that huge emphasis was laid on experiments rather than developing the formulation. It is my opinion that the proposed method is an obvious combination of two traditional NAS methods and I do not see a significant novel contribution in terms of search space design or search strategy. Having said this, I appreciate the authors for the exhaustive and detailed experiments conducted that provide a contemporary review of Weight Sharing and Weight Entanglement in NAS. The authors have included LLM in their study which is a positive sign, however, a discussion on the adverse effect of high computational resources required for NAS on Generative AI would have benefitted the readers. Perhaps a discussion on how the current method be modified to avoid the (re)training of massive (pre-trained) Generative AI might have provided a broader scope for the method.

While the research community and AI/ML practitioners can still perform NAS on traditional AI, the same cannot be said for Generative AI. Thus, the addition of Generative AI as a case study in the work appears to be ornamental. Instead of this, if the authors would have added works with NAS on complex Vision tasks such as Image Enhancement (rather than classification alone), it would have shed light on some of the topics in NAS that remain less explored.

The conclusions are ambitious. While the authors claim that with their proposed approach, consumes less energy, they don't show any evidence for the same. The fact that high computational resources are required by black box methods such as evolutionary algorithms for performing NAS is established over a long time. Comparing the proposed approach with such methods (in their vanilla form) and claiming a large amount of energy saving may not be beneficial and does not signify anything new to the readers working in this domain. Further a marginal improvement in resource-requirement compared to existing gradient based NAS methods will not help researchers and practitioners to eliminate manual tuning completely.

Nevertheless, the work presented will provide a significant starting point for researchers in the domain of NAS and I appreciate for conducting a thorough study with the given idea.

**Potential Impact On The Field Of Automl:**

The work is primarily combining two traditional approaches in NAS in an obvious manner. The amount of novelty is less. However, the work addresses an issue that has not been previously discussed in depth. Hence the work is a good contribution. The researchers working in the domain of NAS will find this work useful and cite it.

**Reproducibility:**

The repository is clear and the work is reproducible.

**Review Confidence:**

5

**Review Rating:**

7

**Review Summary:**

The contribution by the authors is a very thorough study on the proposed idea of combining Weight Entanglement with Gradient based NAS. While the approach to do the same is not entire novel, the experiments conducted will provide several insights to the researchers working in the domain. The only minor flaw that I see is the lack of any new search strategy or method for search space design and a combination of existing methods.

**Technical Quality And Correctness:**

An exhaustive list of experiments are conducted. However, like many NAS works, the present work focusses on classification task alone. While most of the architectures are modular, inclusion of such architectures and tasks such as denoising might have been new. However, the authors have included an LLM. The problem with any form of NAS on LLM is that it is not scalable to larger models. Rather if they would have restricted to a variety of domains in traditional AI, it would have been very beneficial. The work is technically sound.

---

### Official Review · Reviewer_9Dtf · 2024-03-28

**Potential Impact On The Field Of Automl:** Green AutoML
**Potential Impact On The Field Of Automl Rating:** 3
**Technical Quality And Correctness:** The approach and experimentation are …
**Technical Quality And Correctness Rating:** 3
**Clarity Rating:** 4

**Summary Of Contributions:**

This paper proposes a novel scheme to adapt gradient-based methods for weight-entangled spaces.The findings highlight that this approach combines the advantages of gradient-based methods with the memory efficiency of weight-entangled spaces, bridging the gap between these two sub-communities in NAS research.

**Actions Required To Increase Overall Recommendation:**

Please take into account my previous comments and consider implementing them to enhance the paper.

**Clarity:**

Ingeneral, the paper is well-written and easy-to-follow. I have a few minor suggestions that could enhance the clarity of the paper:
- Increasing the size of texts in Fig. 1, Fig. 2, Fig. 4, Fig. 6.
- Figure 4: figures are stretched horizontally.

**Overall Review:**

# Strengths:
- Paper is well-written and addresses important research questions.
- TangleNAS is evaluated on a diverse set of weight-entangled search spaces and tasks including language models.

**Review Confidence:**

4

**Review Rating:**

8

**Review Summary:**

Minor modifications are required for the paper to be considered finalized.

---

### Meta-Review · Area_Chair_q19c · 2024-04-22

**Paper Recommendation:** Accept
**Confidence:** 5

**Metareview:**

All reviewers agree with the contribution of this work. Authors have also carefully prepared the rebuttal and addressed most of the issues mentioned in the review.

---

### Decision · Program_Chairs · 2024-04-29

**Decision:**

Accept

**Comment:**

Thank you for submitting your paper. We are happy to tell you that we accept your paper to the main track. See you in Paris.